# Tunable Perfect Narrow-Band Absorber Based on a Metal-Dielectric-Metal Structure

**Qiang Li [1]**, **Zizheng Li [1,\*]**, **Xiangjun Xiang [2]**, **Tongtong Wang [1]**, **Haigui Yang [1]**, **Xiaoyi Wang [1]**, **Yan Gong [3,4,\*]** and **Jinsong Gao [1,4]**

[1] Key Laboratory of Optical System Advanced Manufacturing Technology, Changchun Institute of Optics, Fine Mechanics and Physics, Chinese Academy of Sciences, Changchun 130033, China; liqiang@ciomp.ac.cn (Q.L.); wangtt@ciomp.ac.cn (T.W.); yanghg@ciomp.ac.cn (H.Y.); wangxiaoyi@ciomp.ac.cn (X.W.); gaojs@ciomp.ac.cn (J.G.)

[2] Research Center of Laser Fusion, China Academy of Engineering Physics, Mianyang 621900, China; dennis55555@163.com

[3] Jiangsu Key Laboratory of Medical Optics, Suzhou Institute of Biomedical Engineering and Technology, Chinese Academy of Sciences, Suzhou 215163, China

[4] College of Da Heng, University of Chinese Academy of Sciences, Beijing 100049, China

\* Correspondence: lizizheng@ciomp.ac.cn (Z.L.); gongy@sklao.ac.cn (Y.G.)

**Abstract:** In this paper, a metal-dielectric-metal structure based on a Fabry–Perot cavity was proposed, which can provide near 100% perfect narrow-band absorption. The lossy ultrathin silver film was used as the top layer spaced by a lossless silicon oxide layer from the bottom silver mirror. We demonstrated a narrow bandwidth of 20 nm with 99.37% maximum absorption and the absorption peaks can be tuned by altering the thickness of the middle $SiO_2$ layer. In addition, we established a deep understanding of the physics mechanism, which provides a new perspective in designing such a narrow-band perfect absorber. The proposed absorber can be easily fabricated by the mature thin film technology independent of any nano structure, which make it an appropriate candidate for photodetectors, sensing, and spectroscopy.

**Keywords:** thin film; coatings; metal-dielectric-metal structure; Fabry–Perot cavity; perfect absorption

## 1. Introduction

Perfect absorbers are of great interest with respect to both fundamental theory and practical applications in many fields, such as solar cells [1–5], sensing [6–10], photo-detection [11,12], and thermal emitting [13,14]. Perfect absorbers can absorb all incident electromagnetic radiation at the desired wavelength, which means reflection and transmission are efficiently suppressed. In recent years, plasmonic structures have attracted much attention due to their excellent ability to achieve the light and matter interaction in infinitesimal space [15–20], by which specific reflection, transmission, and absorption can be achieved. Different kinds of nano structures, such as nanoparticles [21–23], nanocones [24,25], nanohole arrays [26–30], and nano gratings [31–33] were proposed, which can excite surface plasmons resulting in the enhancement of absorption. Many perfect absorbers using nano structures have demonstrated ultra-high absorption and their absorption spectra can be engineered by adjusting the geometry, size, and periodicity of the structures at the same time. However, the fabrication of these nanostructures usually involves costly precise nano fabrication processing steps, such as focused ion beam (FIB) and electron beam lithography (EBL), which turn out to be a challenge for applications in large areas of these patterned absorbers. This is one of the main factors severely limiting their application.

Compared to plasmonic structures, unpatterned thin films made up of one or more films of dielectric or metallic materials are widely used as color filters [34,35], antireflective coatings [36,37], and reflector mirrors [38,39]. Many traditional optical coatings rely on the Fabry–Perot interference effect to achieve a specific function. As early as in 1952, a triple layer consisting of a metal substrate, a dielectric layer, and a thin top metal layer, was proposed as a perfect absorber for radar waves [40]. Recently, Capasso et al. proposed a simple thin film as a perfect absorber, which consists of an ultrathin lossy semiconductor film and a gold reflective mirror [41]. This structure can achieve the interference effect in an ultrathin absorbing film with a thickness of a few nanometers. Unlike optical coatings, the transmission and reflection phase change at the interface are not zero or $\pi$ anymore. The imaginary part of the complex refractive index has a critical impact on the phase change leading to the strong Fabry–Perot-type interference effect, which can form a dip in reflection spectrum. Moreover, thin films can be easily fabricated by mature thin film technology instead of costly nanofabrication techniques, which makes it a candidate in solar cells, photodetectors, and filters. This work has had important research significance and since then many studies have achieved perfect absorption using ultrathin films [42–46]. However, they usually focus on broadband perfect absorption.

In this paper, we propose a Ag-SiO$_2$-Ag (Metal-Dielectric-Metal, MDM) triple layer structure, which can exhibit perfect absorption in the visible and near infrared region with a narrow band. This kind of triple layer structure forms a FP cavity, and its optical spectrum is controlled easily by the thickness of the layers rather than the sub-wavelength size of the nanostructures, which makes it a more compatible candidate in many applications. Moreover, a deep comprehension of the physical mechanism is also studied by the theoretical analysis and finite-different time-domain (FDTD) algorithm, which paves the way to design this kind of perfect absorber. Therefore, the proposed perfect narrow band absorber, with the advantages of low-cost, large area, and a simple fabrication process, is promising for photodetectors, photovoltaics, sensing, and spectroscopy.

## 2. Simulation and Analysis

The proposed triple layer structure is schematically shown in Figure 1a and consists of a bottom Ag layer, a middle SiO$_2$ layer, and a top ultrathin silver (Ag) layer in sequence. In this metal-dielectric-metal (MDM) structure, Ag is chosen for its high reflection in the visible and near infrared region and low material loss. SiO$_2$ is selected for its stable characteristic and appreciable transparency in the specific band. We choose glass as the substrate and the thickness of the bottom Ag layer is 100 nm so that no incident light can pass through. Therefore, the absorption $A$ is equal to $1 - R$ where $R$ is the reflectance. The thickness of the middle SiO$_2$ layer and the top ultrathin Ag layer are set to $d$ and $t$. The 3D finite-difference time-domain (FDTD) algorithm is used to simulate the optical properties of the proposed MDM structure, where the perfectly matched layers (PML) are applied in the $Z$ axis and the periodic boundary conditions are used for a unit cell in the $X$-$Y$ plane. At the same time a 2 nm $\times$ 2 nm $\times$ 2 nm discrete mesh is used for the simulation region. The Ag and SiO$_2$ permittivity in our simulation are from the Palik data [47]. The incident light is a plane wave propagating along the negative $Z$ direction with the wavelength ranging from 400 to 2000 nm.

As we have described before, a FP cavity consisting of two Ag layers separated by a dielectric SiO$_2$ spacer can form resonance when the Bragg principle is met. The thickness of the SiO$_2$ has a significant influence for controlling the absorption peaks at the desired wavelength. The FDTD simulation results in Figure 1b show that high order absorption peaks will appear (the value of $m$ is the resonance mode order) because of the FP resonance as the SiO$_2$ thickness increases. There are six resonance modes corresponding to six narrow absorption bands when the thickness of SiO$_2$ increases from 50 to 900 nm. In the simulated results shown in Figure 1c four absorption peaks are observed at wavelengths of 480.0, 633.0, 941.0, and 1864.0 nm, with a full width at half maximum (FWHM) of 8, 12, 7, and 14 nm when the thickness of the top Ag and SiO$_2$ layers are 30 and 600 nm, respectively. At the same time, the corresponding near electric field maps in Figure 1d confirmed that $m$ order FP resonance was generated in the SiO$_2$ cavity at the wavelength of the absorption peaks in Figure 1b. Moreover, the largest electric

field enhancement for modes from $m = 1$ to $m = 6$ are 5, 6, 14, 22, 12, and 8, respectively, while the electric field enhancement in the air above the top ultrathin Ag layer is much weaker than it in the $SiO_2$ cavity. The proposed MDM structure can achieve a narrowband absorption and the absorption peaks can shift rapidly when the $SiO_2$ thickness changes, and this characteristic benefits a large number of potential applications, such as color filters, bolometers, and sensors.

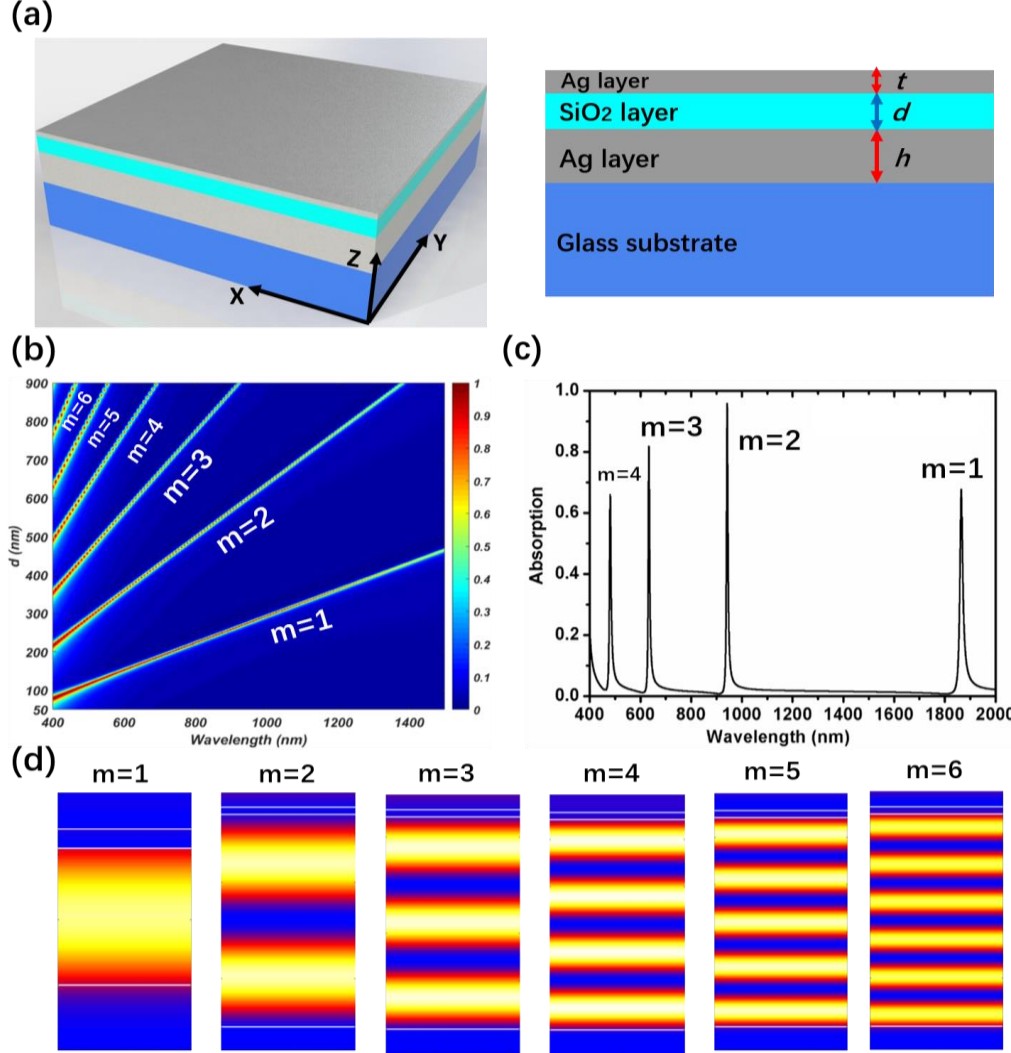

**Figure 1.** (**a**) Schematic diagram of the MDM structure perfect absorber. The thickness of the three layers are $t$, $d$, and $h$, respectively; (**b**) The relationship between the absorption and the $SiO_2$ thickness $d$ by FDTD simulation when the thickness of the top Ag layer is 30 nm; (**c**) Simulated absorption spectra extracted from Figure 1b when $d$ is 600 nm; (**d**) Electric field distribution in the $SiO_2$ cavity at the absorption peaks from $m = 1$ to $m = 6$.

In the previous section, we proposed a three-layer MDM structure forming an asymmetric Fabry−Perot cavity which can achieve multi-narrowband perfect absorption. Now we take the first order of FP resonance ($m = 1$) for example to analyze the optical properties of the triple layer structure and the physical mechanism of the perfect absorption with an ultra-narrow band. Firstly, the transfer matrix method (TMM) is carried out to study the effect of the middle $SiO_2$ thickness on the absorption spectrum. As discussed before, the bottom and top Ag thin films can form a FP resonator cavity and the thickness of the $SiO_2$ spacing layer determines the resonance wavelength. The reflection of the triple MDM structure was calculated using the TMM algorithm with varying $SiO_2$ thickness $d$ from 80 to 200 nm in a 5 nm interval when the thickness of the top ultrathin Ag layer $t$ was fixed at

30 nm, as shown in Figure 2a. The calculated absorption using $A = 1 - R$ is shown in Figure 2b, which exhibits a tunable narrow band perfect absorption in the visible region. In addition, we identified the relationship of the resonance wavelength and the $SiO_2$ cavity length by the minimum value of the calculated reflection curves, as shown in Figure 2c, in which we can see there is a linear relationship between the resonance wavelength and thickness $d$. We choose five different $SiO_2$ thicknesses—90, 110, 130, 160, and 180—and simulated the absorption spectrum by the FDTD algorithm, as shown in Figure 2d. The absorption spectra calculated by TMM in Figure 2c and the FDTD algorithm in Figure 2d agree well with each other, which can verify the correctness of the calculation and simulation.

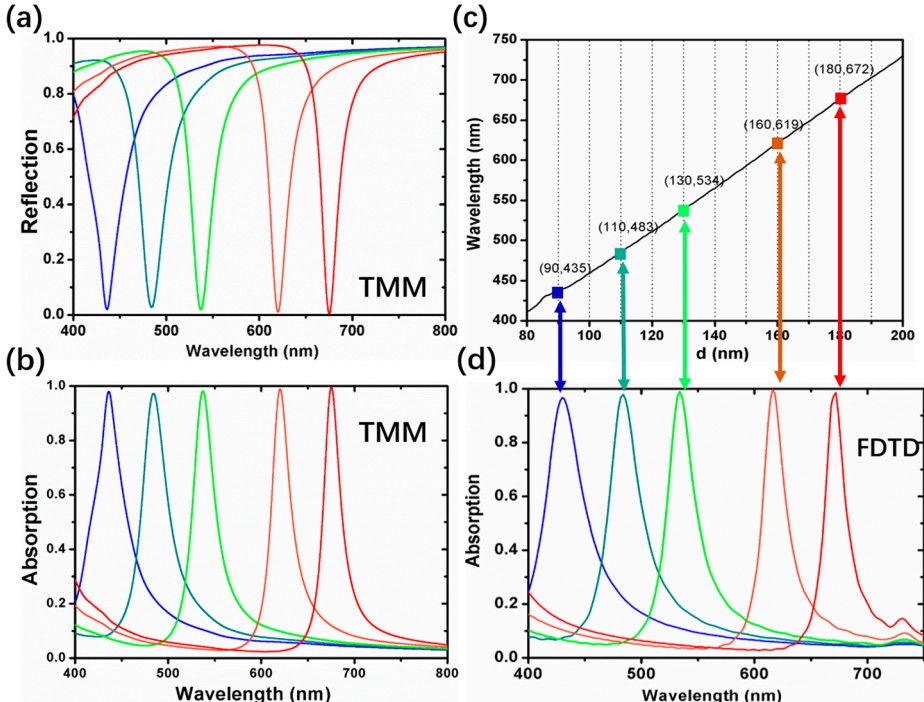

**Figure 2.** (**a**,**b**) The calculated reflection and absorption of five $SiO_2$ thickness $d$ (90, 110, 130, 160, and 180 nm) by the TMM algorithm. (**c**) The relationship of resonance wavelength and the $SiO_2$ thickness $d$ calculated by the TMM algorithm; (**d**) The five colorful solid lines represent the simulated absorption spectra by the FDTD algorithm when the thickness of the $SiO_2$ $d$ are 90, 110, 130, 160, and 180 nm, respectively.

In order to clarify the physical mechanism of the narrowband perfect absorber, the effect of the different top Ag layer thickness $t$ on the absorption properties is discussed. The FDTD algorithm was used to simulate the reflection and transmittance of a single Ag layer by changing the thickness from 10 to 70 nm on a glass substrate, and the calculated absorption is shown in Figure 3a. As we can see, the absorption mainly concentrates in the short wavelength region, which is determined by the property of the Ag material. Figure 3b shows the absorption curves when the thickness of the Ag layer is 10, 30, and 50 nm, respectively. The absorption is relatively low so as no resonant behavior is observed. Compared with the single Ag layer, the MDM triple layer structure shows obviously resonant absorption, as shown in Figure 3c (here the middle $SiO_2$ thickness is fixed to be 130 nm). If there is no top Ag silver film, the structure turns out to be an Ag reflection mirror coated by a $SiO_2$ layer, which is why the reflection is ultrahigh leading to almost zero absorption when $t$ is 0 nm. As the thickness of the top Ag layer increases, an obvious resonant absorption band can be generated due to the FP cavity. For a 30 nm thickness top Ag layer the resonance wavelength is approximately 534 nm and the absorption is higher than other thicknesses, as shown in Figure 3d. Although the bandwidth is narrower when the thickness $t$ is 50 nm, its absorption declines dramatically. Therefore, the ultrathin top Ag layer in our work is optimized at 30 nm for perfect absorption.

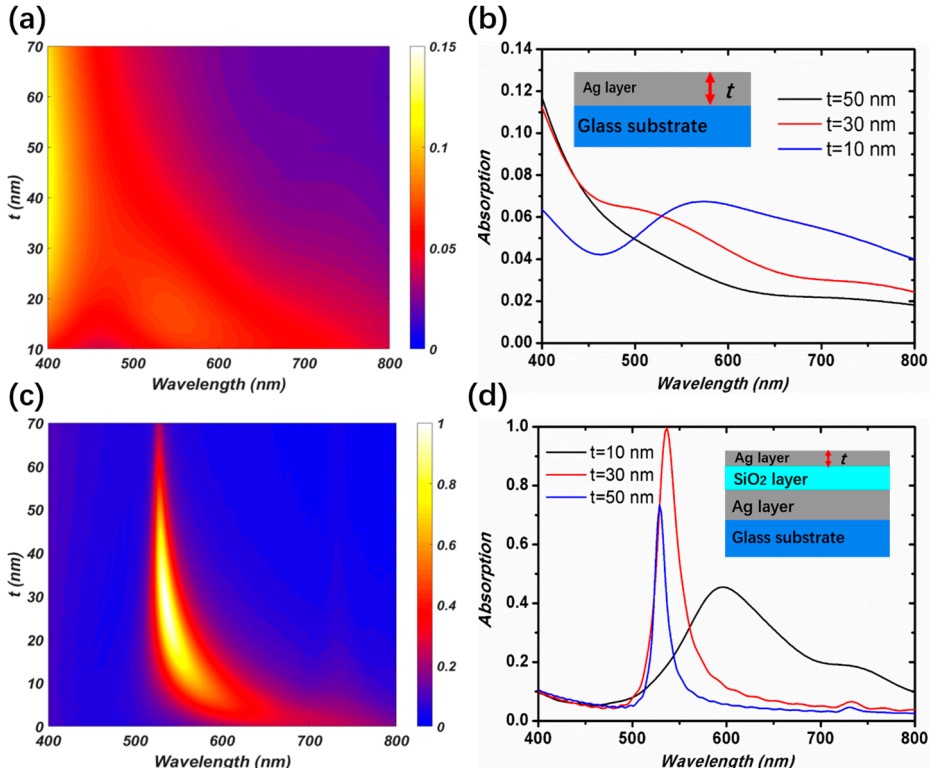

**Figure 3.** The simulated absorption by the FDTD algorithm. (**a**) The absorption curves when a single Ag layer coated on the glass substrate with varying thickness from 0–70 nm; (**b**) The absorption curves taken from Figure 3a when the thickness of the top Ag layer is 10, 30 and 50 nm; (**c**) The absorption as a function of wavelength of the MDM structure with a top Ag thickness from 0–70 nm when the middle SiO$_2$ thickness is fixed at 130 nm; (**d**) The absorption curves taken from Figure 3c when the thickness of the top Ag layer is 10, 30 and 50 nm.

In addition, the distributions of the FDTD simulated electric field (|E|) at wavelengths of 534 nm (FP resonance) and 700 nm (no FP resonance) for the triple layer MDM structure with 30 nm top Ag and 130 nm middle SiO$_2$ layers are shown in Figure 4a,b. As we can see, the electric field is enhanced in the air region above the ultrathin top Ag layer and in the middle SiO$_2$ cavity there is a small E-field at the non-resonance wavelength (700 nm) of the FP cavity, while for the resonance wavelength of 534 nm, we can observe that the E-field is mainly concentrated in the SiO$_2$ cavity because of the FP resonance, which leads to a four times enhancement in the E-field intensity. Figure 4c shows the distributions of the E-field in the visible region. According with Figure 4b, the E-field intensity is obviously enhanced at the resonance wavelength of 534 nm and the E-field enhancement band is narrow leading to a narrow absorption band. The FDTD simulation results indicate that the proposed triple MDM structure realizes the narrow band absorption via the FP cavity, and the perfect absorption has a relation with the E-field enhancement in the middle SiO$_2$ cavity.

In order to clarify the relation of perfect absorption and the E-field enhancement in the cavity, we calculated the absorption of three layers, respectively, using the FDTD algorithm for the MDM structure with a 30 nm top Ag layer and a 130 nm thickness middle SiO$_2$ layer when the FP resonance occurs in the cavity at the wavelength of 534 nm. The power monitors were set from the bottom Ag layer to the ultrathin top layer Ag layer with a 1 nm interval to record the power intensity if the incident light passed through them. The difference values between two neighboring monitors can show the absorption in this 1 nm region. The simulated result shows in Figure 5a that incident light is completely absorbed by two Ag layers because the SiO$_2$ material has no absorption property. The incident light power is mainly absorbed by the top ultrathin Ag layer. We also obtained the absorption in the bottom

and top Ag layers as about 62.75% and 36.28%, respectively. The remaining 1% of incident light is reflected back, which is in accordance with the simulated absorption curve in Figure 2b.

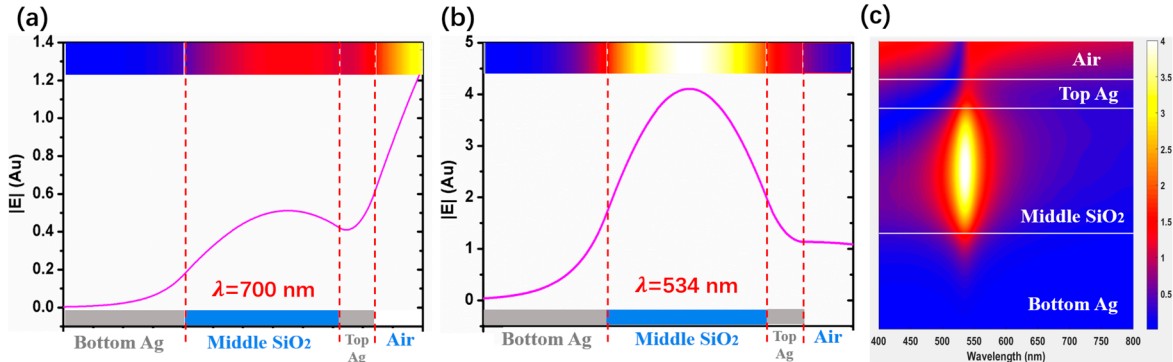

**Figure 4.** (**a**,**b**) Simulated E-field intensity distributions by the FDTD algorithm at the resonance wavelength of 534 nm and non-resonance wavelength of 700 nm, respectively. The three layers of the MDM absorber consist of a 30 nm top Ag layer, a 130 nm middle SiO$_2$ layer, and a 100 nm bottom Ag layer. (**c**) Simulated E-field distributions as a function of wavelength in the visible region.

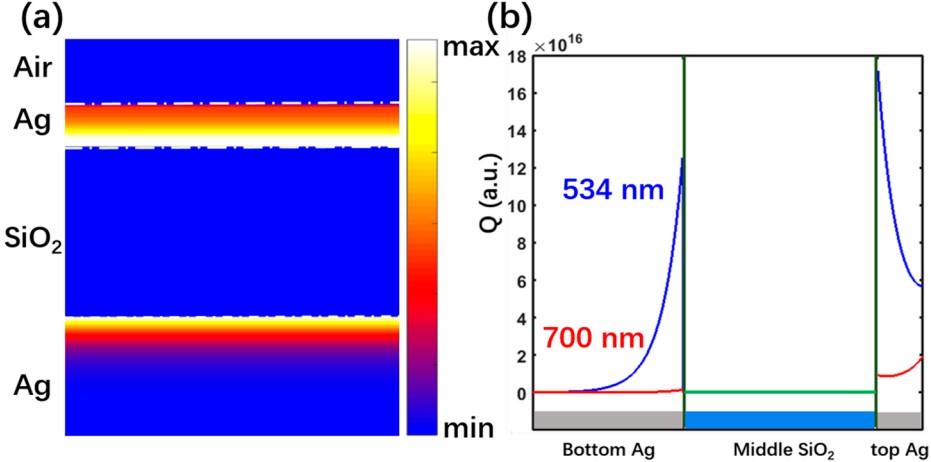

**Figure 5.** (**a**) Simulated energy distribution at the 534 nm resonance wavelength when the middle SiO$_2$ layer is 130 nm; and (**b**) the Ohmic loss calculated by Equation (1) at the resonance wavelength and non-resonance wavelength.

Furthermore, the absorption distribution in the proposed triple layers can be directly calculated by the local Ohmic loss formula [48]:

$$A(r, w) = \frac{1}{2}\varepsilon_0 w Im\varepsilon(w)|E|^2 \tag{1}$$

in which $w$ is the angular frequency, $Im\varepsilon(w)$ is the imaginary part of the dielectric permittivity, and E is the simulated E-field by the FDTD algorithm, as shown in Figure 4. In this way, we calculated the absorption distribution in the triple layer stack using Equation (1), as plotted in Figure 5b. As we can see, the calculated Ohmic loss at the resonance wavelength and non-resonance wavelength is completely different. The blue line indicated the Ohmic loss at resonance wavelength is much larger than the red line which is indicated the Ohmic loss at non-resonance wavelength. The Ohmic loss totally occurs in both bottom and top metal Ag layers, and the ultrathin top Ag layer plays a dominating role, which is in accordance with the simulated results in Figure 5a. Obviously, the proposed triple

layer perfect absorber is based on the FP cavity, and the thickness of the middle SiO$_2$ determines the wavelength of resonance. When the FP resonance formed in the cavity, the selected incident light can be reflected constantly by the two Ag layers, and some incident energy is transformed to ohmic loss each time light is reflected until all the power is consumed. The stronger E-field in the metal, the more power it can consume, which lays the foundation of the proposed perfect absorber.

## 3. Experiment Results and Discussion

We used the method of magnetron sputtering to fabricate the triple layer structure. Ag and SiO$_2$ layers were deposited alternately on a glass substrate via DC and RF sputtering in a vacuum chamber (SKY-450, Sky Technology Development, Shenyang, China). The deposition rate of Ag and SiO$_2$ are 4.28 and 2.63 Å/s, respectively. After research of the fabrication technology, the detailed deposition parameters are shown as Table 1 below.

**Table 1.** Coating process parameters.

| Material | Power (W) | Ar (Sccm) | O$_2$ (Sccm) | Vacuum Degree (Pa) | Deposition Rate (Å/s) |
|---|---|---|---|---|---|
| Ag | 150 | 80 | 0 | 1.0 | 4.28 |
| SiO$_2$ | 200 | 80 | 20 | 1.5 | 2.63 |

The optical character of a single SiO$_2$ layer has an important influence on our proposed perfect absorber. We use a spectroscopic ellipsometer to measure the optical constant of the SiO$_2$ and Ag thin film fabricated by the magnetron sputtering method. The measured results are shown in Figure 6, which illustrates that the deposition parameters we chose are appropriate.

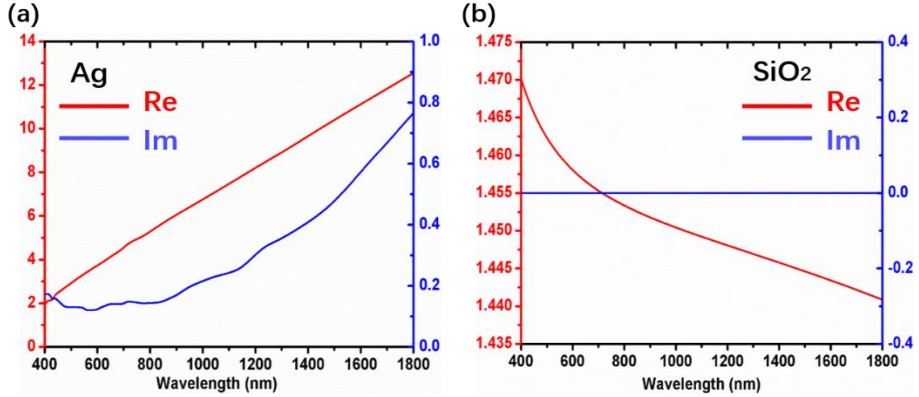

**Figure 6.** The optical constant of (**a**) Ag and (**b**) SiO$_2$ from the results measured by a spectroscopic ellipsometer.

After studying the process, a sample with a 100 nm bottom Ag layer, a 600 nm middle SiO$_2$ layer, and a 30 nm ultrathin top layer was fabricated. Its reflection was measured by a spectrometer (Perkin Elmer Lambda 900, Waltham, MA, USA), and the absorption calculated by $A = 1 - R$, as shown in Figure 7. Three resonance absorption peaks corresponding to 4, 3, and 2 order FP cavities can be observed at the wavelengths of 480.0, 633.0, and 941.0 nm. The absorption bands are narrow and agree perfectly with the FDTD simulation results, which can verify the correctness of the FDTD simulation.

In addition, we achieved narrow-band perfect absorption in the visible region using the first order of FP resonance. The fabricated samples with five different middle SiO$_2$ thicknesses present vivid colors, from left to right, as Figure 8a shows. The scanning electron microscope (SEM, JMS-6510, JEOL, Tokyo, Japan) image shows that the SiO$_2$ and Ag layers are clearly seen in Figure 8b. The reflection and absorption spectra obtained by FDTD simulation (the solid lines) and experiment (the dotted lines) are shown in Figure 8c, from which we can see the absorption can reach above 99.37% and their full width

at half maximum (FWHM) are about 20–50 nm leading to a highest quality factor of 35.7. The results obtained by experiment compared with FDTD simulation are shown in detail in Table 2.

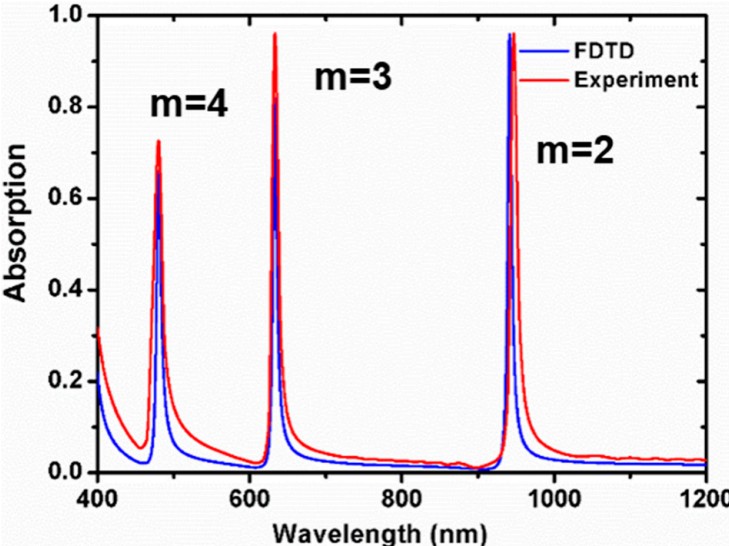

**Figure 7.** Simulated and experimental absorption for the 100 nm bottom Ag layer, 600 nm middle $SiO_2$ layer, and 30 nm top Ag layer.

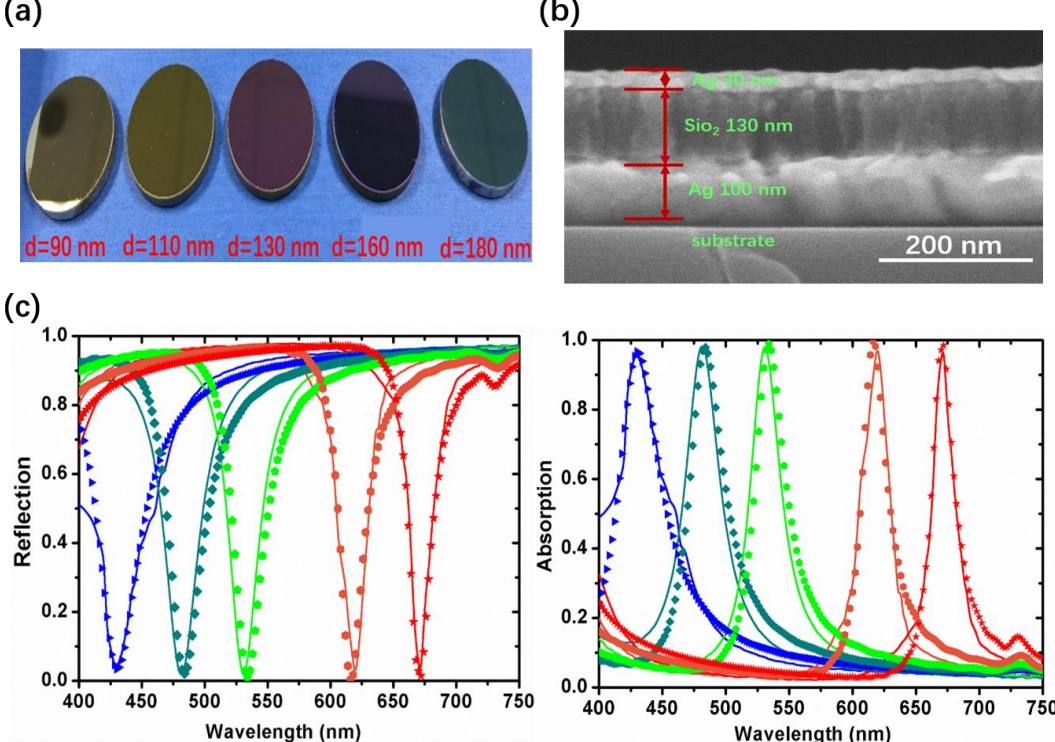

**Figure 8.** (**a**) Five samples with different $SiO_2$ thickness: 90, 110, 130, 160, and 180 nm, from left to right; (**b**) SEM image of the structure showing the Ag and $SiO_2$ layers; (**c**) Simulated and experimental reflection and absorption of five samples; The solid lines are the FDTD simulation results, while the dotted lines are the experimental results.

**Table 2.** The experiment and simulation results at resonance wavelengths for five different thicknesses.

| *d* (nm) | Simulated Resonance Wavelength (nm) | Simulated Max Absorption | Experimental Resonance Wavelength (nm) | Experimental Max Absorption | FWHM (nm) |
|---|---|---|---|---|---|
| 90 | 436 | 0.9779 | 440 | 0.9868 | 54 |
| 110 | 485 | 0.9887 | 482 | 0.9865 | 30 |
| 130 | 534 | 0.9899 | 530 | 0.9937 | 26 |
| 160 | 617 | 0.9966 | 620 | 0.9925 | 24 |
| 180 | 671 | 0.9984 | 675 | 0.9917 | 20 |

## 4. Conclusions

In conclusion, we propose a Ag-SiO$_2$-Ag triple layer structure, which can exhibit perfect absorption in the visible and near infrared region with a narrow band based on the Fabry−Perot cavity. Its optical spectrum is controlled easily by the thickness of the layers and the highest experimental absorption can reach about 99.37% with a narrow bandwidth of 20–50 nm. Moreover, a deep comprehension of the physical mechanism is also studied by theoretical analysis and the FDTD algorithm, which paves the way to design this kind of triple structure. The function of each layer and the physical mechanism in the proposed MDM triple layers are clarified using FDTD simulation. This kind of perfect absorber can be fabricated using mature deposition film technology with the advantages of low-cost, large area, and a simple fabrication process, which make it a promising solution for photodetectors, photovoltaics, sensing, and spectroscopy.

**Author Contributions:** Conceptualization: Q.L. and Z.L.; Methodology: Q.L. and Z.L.; Validation: Q.L. and X.W.; Formal Analysis: Q.L. and H.Y.; Investigation: T.W., X.X. and X.W.; Data Curation: Q.L., Y.G. and J.G.; Writing—Original Draft Preparation: Q.L.; Writing—Review and Editing: H.Y., Q.L. and Z.L.

**Funding:** This research was funded by the National Natural Science Foundation of China (Nos. 61705226 and 61875193), the Changchun Science and Technology Innovation "Shuangshi Project" Major Scientific and Technological Project (No. 19SS004), and the Science and Technology Innovation Project of Jilin Province (No. 20190201126JC).

**Conflicts of Interest:** The authors declare no conflict of interest.

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
