# Peer review of "Tunable Perfect Narrow-Band Absorber Based on a Metal-Dielectric-Metal Structure"

_coatings, doi:10.3390/coatings9060393_

Round 1

Reviewer 1 Report

In this article, the authors propose a triple Ag-SiO2-Ag layer structure, which can exhibit

absorption in the visible and near infrared region with a narrow band based on Fabry−Perot cavity.

1)      The authors didn’t discuss the novelty of their work in comparison to previous research works on metal/dielectric structures. See for example:

Chaorong Li et al 2010 Nanotechnology 21 245602

Y.C. Lin, Z.A. Chen, and C.H. Shen, Phys. Procedia 32, 19 (2012)

 Micro-Raman investigation of Ag/graphene oxide/Au sandwich structure, Mater. Res. Express 6 075605

 Graphene oxide on magnetron sputtered silver thin films for SERS and metamaterial applications,

Applied Surface Science, 427 (2018)

2)      The authors should add the plots of the magnetic field distributions at the resonance wavelength.

3)      The authors should discuss the LSPs modes supported by their MDM structure.

Author Response

Please see the PDF file.

Reviewer 2 Report

 In this paper, authors have successfully fabricated the tunable perfect narrow-band absorber based on metal-dielectric-metal structure. Optical simulation results and experimental results agree well. However, it seems that authors should show more the novelty of this work, compared to the conventional perfect absorbers.  

1. There have been already many reports on metal-oxide-metal absorber including Ag-SiO2-Ag. What is the new concept of this work? If authors think this work has an enough novelty, authors should explain more about that, comparing with the conventional perfect absorbers; Wu et al., Nanoscale Research Letters 2017 12:427, Liao et al., Optics express 25.25 (2017): 32080-32089. Zhao et al., Optics express 25.8 (2017): A208-A222.

2. The description of the experimental part is insufficient. For metal oxides, the optical property generally differs with regard to the sputtering conditions such as partial oxygen pressures. Therefore, authors should clarify the optical property of single SiO2 thin film. Furthermore, what is the maximum temperature for the perfect absorber?

3. Authors should clarify the crystal morphology of the fabricated SiO2 layer; amorphous phase or polycrystalline?

4. In general, very thin silver film is mechanically weak; the film is easily removed or scratched by mechanical force. Author should comment on this point.

Author Response

Please see the PDF file.

Round 2

Reviewer 1 Report

The authors have addressed some of my concerns. However, I think that the authors should improve the english language of the article and the introduction must be improved. In fact, the introduction doesn't  provide sufficient background and adeguate references. Furthermore, I would suggest the authors to make a comparison between the present work and the previous work (

[1]         Q. Li, J. Gao, H. Yang, H. Liu, X. Wang, Z. Li, X. Guo, Tunable Plasmonic Absorber Based on Propagating and Localized Surface Plasmons Using Metal-Dielectric-Metal Structure, Plasmonics. 12 (2017) 1037–1043. doi:10.1007/s11468-016-0356-5.

), which describes a metal -dielectric-metal structure, as in present article. 

Reviewer 2 Report

 Authors have replied to all my concerns and those were well reflected in the revised manuscript. I suggest the publication of this manuscript with no further revision.